# MergeBERT: Program Merge Conflict Resolution via Neural Transformers

## Abstract

Collaborative software development is an integral part of the modern software development life cycle, essential to the success of large-scale software projects. When multiple developers make concurrent changes around the same lines of code, a merge conflict may occur. Such conflicts stall pull requests and continuous integration pipelines for hours to several days, seriously hurting developer productivity.

In this paper, we introduce MergeBERT, a novel neural program merge framework based on the token-level three-way differencing and a transformer encoder model. Exploiting restricted nature of merge conflict resolutions, we reformulate the task of generating the resolution sequence as a classification task over a set of primitive merge patterns extracted from real-world merge commit data.

Our model achieves 63–68% accuracy of merge resolution synthesis, yielding nearly a 3× performance improvement over existing structured, and 2× improvement over neural program merge tools. Finally, we demonstrate that MergeBERT is sufficiently flexible to work with source code files in Java, JavaScript, TypeScript, and C# programming languages, and can generalize zero-shot to unseen languages.

## 1 Introduction

Collaborative software development relies on version control systems such as `git` to track changes across files. In most projects, developers work primarily in a branch of a software repository, periodically synchronizing their code changes with the `main` branch via pull requests (Gousios et al., 2016). When multiple developers make concurrent changes to the same line of code, a merge conflict may occur. According to an empirical study of four large software projects by Zimmermann (2007) up to 46% of all merge commits result in conflicts. Resolving merge conflicts is a time-consuming, complicated, and error-prone activity that requires understanding both the syntax and program semantics, often taking more time than developing a code feature itself (Bird & Zimmermann, 2012).

Modern version control systems such as `git` utilize the `diff3` algorithm for performing unstructured line-based three-way merge of input files (Smith, 1998). This algorithm aligns the two-way diffs of two versions of the code $\mathcal{A}$ and $\mathcal{B}$ over the common base $\mathcal{O}$ into a sequence of diff "slots". At each slot, a change from either $\mathcal{A}$ or $\mathcal{B}$ is selected. If both program versions introduce a change at the same slot, a merge conflict is produced, and manual resolution of the conflicting modifications is required.

A versatile, production-level merge conflict resolution system should be aware of programming language syntax and semantics yet be sufficiently flexible to work with any source code files, irrespective of the programming language. It should generalize to a wide variety of real-world merge conflicts beyond a specific merge type or a domain of software artifacts.

Inspired by the exceptional performance of transformer models and self-supervised pretraining in natural language understanding and generation tasks (Devlin et al., 2019; Radford et al., 2019; Liu et al., 2019; Lewis et al., 2019; Raffel et al., 2020) as well as in the programming language domain (Feng et al., 2020; Svyatkovskiy et al., 2020; Clement et al., 2020; Tufano et al., 2020; Ahmad et al., 2021), we introduce MergeBERT: a neural program merge framework based on token-level three-way differencing and transfer learning. We select a bidirectional transformer encoder (BERT) as our encoder implementation. As a bidirectional encoder, BERT allows to attend to code context surrounding the conflicting chunks, which is a key advantage over left-to-right language models. To

endow our model with a basic knowledge of programming language syntax and semantics, we adopt a two-step training procedure: (1) unsupervised masked language model pretraining on a massively multilingual source code corpus, (2) supervised finetuning for the sequence classification task. We transfer weights of the pretrained encoder into a multi-input model architecture that encodes all inputs that a standard `diff3` algorithm takes (two two-way diffs of input programs) as well as the edit sequence information, then aggregate them for learning. While MergeBERT utilizes BERT, other encoder architectures such as LSTM (Hochreiter & Schmidhuber, 1997), or efficient transformer variants like Poolingformer (Zhang et al., 2021) could also be utilized for this task.

The paper contributions are as follows: (1) we introduce MergeBERT a novel transformer-based program merge framework that leverages token-level three-way differencing and formulates the task of generating the resolution sequence as a classification task over a set of primitive merge patterns extracted from real-world merge commit data, (2) we effectively transfer knowledge about program syntax and types of source code identifiers from millions of software programs to downstream sequence classification task of merge conflict resolution by using self-supervised pretraining (see section 5), which also makes this approach computationally more feasible, (3) we overcome several limitations of the existing neural program merge models Dinella et al. (2021) and semi-structured program merge tools like jsFSTMerge and JDime to improve upon the state-of-the-art by 2–3× (see sections 7 and 8), and finally, (4) we demonstrate that multilingual MergeBERT is sufficiently versatile to work with programs in Java, JavaScript, TypeScript, and C#, and can generalize to unseen languages without retraining.

## 1.1 RELATED WORK

There have been multiple attempts to improve merge algorithms by restricting the them to a particular programming language or a specific type of applications (Mens, 2002). Typically, such attempts result in algorithms that do not scale well or have a low coverage.

Syntactic merge algorithms improve upon `diff3` by verifying the syntactic correctness of the merged programs. Several syntactic program merge techniques have been proposed (Westfechtel, 1991; Asklund, 1999) which are based on parse trees or abstract syntax trees and graphs.

Apel *et al.* noted that structured and unstructured merge each has strengths and weaknesses. They developed a semi-structured merge tool, FSTMERGE, which switches between approaches (Apel et al., 2010). They later introduced JDIME, an approach that automatically tunes a mixture of structured and unstructured merge based conflict locations (Apel et al., 2012). AUTOMERGE tool (Zhu & He, 2018) improves upon the JDIME by adding a built-in matcher to check output resolutions for the merges performed by JDIME. AUTOMERGE synthesizes resolutions only when the structured merge algorithm fails to complete a merge. Trindade Tavares et al. (2019) later implemented jsFSTMerge by adapting an off-the-shelf grammar for JavaScript and modifying the FSTMerge algorithm itself to address JavaScript specific issues.

In addition, Pan et al. (2021) explore using program synthesis to learn repeated merge resolutions within a project. However, the approach is limited to a single C++ project, and only deals with restricted cases of import statements. Sousa et al. (2018) explore the use of program verification to certify that a merge obeys a semantic correctness criteria, but does not help resolve merge conflicts.

## 2 MOTIVATING EXAMPLE

In this section, we formulate the traditional line-level merge conflict resolution problem as a classification task and show how token-level formulation of `diff3` helps to localize the merge conflicts. Fig. 1 provides an example merge conflict in JavaScript. Fig. 1(a) on the left shows the standard `diff3` markers "<<<<<<< A.js", "||||||| O.js", "=======" and ">>>>>>> B.js", which denote the conflicting regions introduced by programs $\mathcal{A}$, base $\mathcal{O}$, and $\mathcal{B}$ respectively. Here, $\mathcal{O}$ represents the most common ancestor of programs $\mathcal{A}$ and $\mathcal{B}$ in the version control history. We denote the program text of `diff3` conflicting regions as $A_j$, $B_j$, $O_j$, where $j$ is a conflict index. The conflict index may be omitted when referring to programs consisting of a single conflict only. We refer to the program text outside the conflicting chunks, common to all merged programs versions, as a prefix and suffix, and denote it respectively as `Pref` and `Suff` throughout the paper. First, MergeBERT represents each line-level merge conflict instance at token level (Fig. 1(b)) which localizes conflicting regions,

yielding $a \subset A$, $b \subset B$, $o \subset O$, and then it invokes the neural model to predict a resolution via classification (Fig. 1(c)). Here, and throughout the paper we will use lower case notations to refer to attributes of token-level differencing (e.g. $a$, $b$, and $o$ are conflict regions produced by `diff3` of token granularity). Intuitively, token `diff3` first turns the line-structured text into a list of tokens (including space and line delimiters), applies standard `diff3` algorithm to the resulting documents, and reconstructs the merged document at line level. As a result of token-level merge, the whole "`let x = max(y,`" string is cleanly merged, becoming a part of the program prefix, and "`)`" is prepended to the program suffix.

Token-level `diff3` is a syntactic merge algorithm. As such, it does not guarantee semantic correctness of the merged program. In most cases (82%), however, token merge matches the user resolution.

Observe, that the resolution does not consist of any single line from either $A$ or $B$ since both edits modify a common line present in the base. Hence, earlier neural approaches such as DeepMerge (Dinella et al., 2021) would not be able to synthesize the resolution. On the other hand, structured merge techniques (such as jsFSTMergeby Trindade Tavares et al. (2019)) cannot resolve the conflict as the conflict appears on a program statement, which leads to side effects (e.g. syntactically incorrect code).

A token-level merge can interleave edits within lines (i.e., tokens in which one edit does not conflict with another are trivially merged). Consider $\mathcal{A}$'s edit of the `var` to `let` keyword. Such non-conflicting edits suffice to demonstrate the above. Likewise, consider the token-level conflict for the arguments to the `max` function: an appropriate model trained on JavaScript should be able to easily deduce that taking the edit from $\mathcal{B}$ (i.e., "11, z") captures the behavior of $\mathcal{A}$'s edit as well. The suggested resolution gives an intuitive demonstration for how MergeBERT turns a complex line-level resolution into a simpler token-level classification problem.

MergeBERT can deal with non-trivial real-world merges composed of multiple conflicting chunks. We include a complete example of such a merge conflict in the Appendix.

| (a) Line-level conflict | (b) Token-level conflict | (c) Resolved merge |

Figure 1: Example merge conflict represented through standard `diff3` (left) and token-level `diff3` (center), and the user resolution (right). The merge conflict resolution takes the token-level edit $b$.

## 3 BACKGROUND: DATA-DRIVEN MERGE

Dinella et al. (2021) introduced the *data-driven program merge* problem as a supervised machine learning problem. A program merge consists of a 4-tuple of programs $(\mathcal{A}, \mathcal{B}, \mathcal{O}, \mathcal{M})$, where

1. The base program $\mathcal{O}$ is the most common ancestor in the version history for programs $\mathcal{A}$ and $\mathcal{B}$,

2. `diff3` produces an unstructured line-level conflict when applied to $(\mathcal{A}, \mathcal{B}, \mathcal{O})$, and

3. $\mathcal{M}$ is the program with the developer resolution, having no conflicts.

Given a set of such programs and merges $(\mathcal{A}, \mathcal{B}, \mathcal{O}, \mathcal{M})$, the goal of a data-driven merge is to learn a function, merge, that maximizes the set of examples where $\text{merge}(\mathcal{A}, \mathcal{B}, \mathcal{O}) = \mathcal{M}$. Moreover,

since a program may have multiple unstructured conflicts $(A_j, B_j, O_j, M_j)$, j=0...N, the data-driven merge considers the different *merge tuples* corresponding to the conflicting regions *independently*, and poses the learning problem over all the merge tuples present in $(\mathcal{A}, \mathcal{B}, \mathcal{O}, \mathcal{M})$. Dinella et al. (2021) also provide an algorithm for extracting the exact resolution regions for each merge tuple and define a dataset that corresponds to *non-trivial* resolutions where the developer does not drop one of the changes in the resolution. Further, they provide a sequence-to-sequence encoder-decoder based architecture, where a bi-directional gated recurrent unit (GRU) is used for encoding the merge inputs comprising of $(A, B, O)$ segments of a merge tuple, and a *pointer mechanism* is used to restrict the output to only choose from line segments present in the input. Given the restriction on copying only lines from inputs, the dataset defined in the paper did not consider merges where the resolution required token-level interleaving, such as the conflict in Figure 1. And, lastly, the dataset consists of merge conflicts in a single language, JavaScript. In contrast, our paper addresses both of these limitations.

## 4 MERGE CONFLICT RESOLUTION AS A CLASSIFICATION TASK

In this work, we demonstrate how to exploit the restricted nature of merge conflict resolutions (compared to an arbitrary program repair) to leverage discriminative models to perform the task of generating the resolution sequence. We have empirically observed that a `diff3` of token granularity enjoys two useful properties over its line-level counterpart: (i) it helps localize the merge conflicts to small program segments, effectively reducing the size of conflicting regions, and (ii) most resolutions of merge conflicts produced by token `diff3` consist entirely of changes from $a$ or $b$ or $o$ or a sequential composition of $a$ followed by $b$ or vice versa. On the flip side, a token-level merge has the potential to introduce many small conflicts. To balance the trade-off, we start with the line-level conflicts as produced by the standard `diff3` and perform a token-level merge of only the segments present in the line-level conflict. There are several potential outcomes for such a two-level merge at the line-level:

- *A conflict-free token-level merge*: For example, the edit from $A$ about `let` is merged since $B$ does not edit that slot as shown in Fig. 1(b).

- *A single localized token-level merge conflict*: For example, the edit from both $A$ and $B$ for the arguments of `max` yields a single conflict as shown in Fig. 1(b).

- *Multiple token-level conflicts*: Such a case (not illustrated above) can result in several token-level conflicts.

Token-level `diff3` applied to a merge tuple $(\mathcal{A}, \mathcal{B}, \mathcal{O}, \mathcal{M})$, would generally result in a set of localized conflicting regions and resolutions $\langle a_j, b_j, o_j, r_j \rangle_j$. We empirically observe that majority of such $r_j$ are comprised entirely of (i) $a_j$, (ii) $b_j$, (iii) $o_j$, concatenating (iv) $a_j, b_j$ or (v) $b_j, a_j$, and variants of (iv–v) having the lines present in the base excluded, comprising a total of *nine* primitive merge resolution patterns (see section 10.1 in Appendix for more details about the primitive merge patterns).

We, therefore, can treat the problem of constructing $r_j$ as a classification task to predict between these possibilities. It is important to note that although we are predicting simple resolution strategies at token-level, they translate to complex interleavings at line-level. Of course, not all line-level conflicts are resolved by breaking that conflict to tokens—some resolutions which are complex line-based interleavings are not expressible as a choice at the token-level.

The key practical advantage of formulating merge conflict resolution as a classification task is a significant reduction in total FLOPS required to decode a resolution region, as compared to generative models, making this approach an appealing candidate for deployment in IDEs (see section 11.1).

## 5 MERGEBERT: NEURAL PROGRAM MERGE FRAMEWORK

MergeBERT is a textual program merge model based on the bidirectional transformer encoder (BERT) model. It approaches merge conflict resolution as a sequence classification task given conflicting regions extracted with token-level differencing and surrounding code as context. The key technical

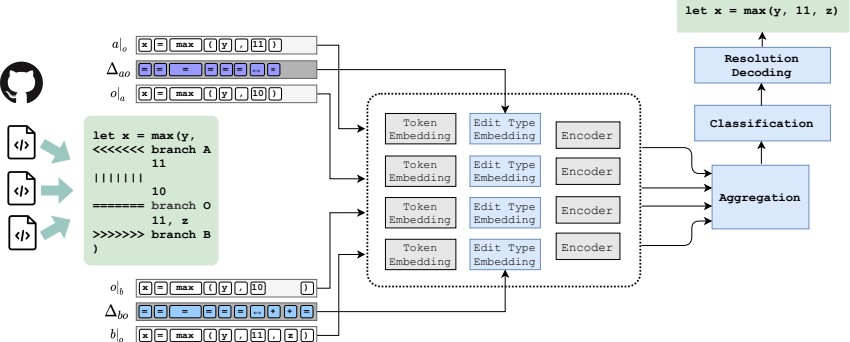

Figure 2: An overview of the MergeBERT architecture. From left to right: given conflicting programs $\mathcal{A}$, $\mathcal{B}$ and $\mathcal{O}$ token-level differencing is performed first, next, programs are tokenized and the corresponding sequences are aligned ($a|_o$ and $o|_a$, $b|_o$, and $o|_b$). We extract edit steps for each pair of token sequences ($\Delta_{ao}$ and $\Delta_{bo}$). Four aligned token sequences are fed to the multi-input encoder neural network, while edit sequences are consumed as type embeddings. Finally, encoded token sequences are summarized into a hidden state which serves as input to classification layer. See Algorithm 1 for details about merge resolution decoding. Parts of the neural networks colored in blue are finetuned, the rest are transferred from pretrained encoder and frozen.

innovation to MergeBERT lies in how it breaks program text into an input representation amenable to training with a transformer encoder and how it aggregates various input encodings for classification.

MergeBERT exploits the traditional two-step pretraining and finetuning training procedure. We use self-supervised masked language modeling (MLM) pretraining on a massively multilingual source code corpus followed by supervised finetuning for a classification task (section 11.4 provides details about the implementation). For finetuning, we construct a multi-input model architecture that encodes pair-wise aligned token sequences of conflicting programs $\mathcal{A}$ and $\mathcal{B}$ with respect to original program $\mathcal{O}$, as well as corresponding edit sequence steps, then aggregate them for learning. An overview of MergeBERT model architecture is shown in Fig. 2.

Given a merge tuple $(\mathcal{A}, \mathcal{B}, \mathcal{O}, \mathcal{M})$ with token-level conflicting chunks $a_j, b_j, o_j$, MergeBERT models the following conditional probability distribution:

$$p(r_j | a_j, b_j, o_j), \tag{1}$$

and consequently, for entire programs:

$$p(R | A, B, O) = \prod_{j=1}^{N} p(r_j | a_j, b_j, o_j) \tag{2}$$

where $N$ is the number of token-level conflicts in the merge tuple $(\mathcal{A}, \mathcal{B}, \mathcal{O}, \mathcal{M})$.

Treating token-level conflicts independently is a simplifying assumption. However, data shows that only 5% of merge conflicts result in more than 1 token-level conflict per line-level conflict.

## 5.1 REPRESENTING MERGE CONFLICTS

As shown by Dinella et al. (2021), an effective merge representation needs to be "edit aware" to provide an indication that $A$ and $B$ are edits of the original program $O$. Prior work on distributed representations of edits (Yin et al., 2019) describes how to compute a two-way diff using a standard deterministic diffing algorithm and represent the resulting pair-wise alignment as a vector consumable by machine learning models.

Given a merge tuple $(\mathcal{A}, \mathcal{B}, \mathcal{O}, \mathcal{M})$, MergeBERT first calculates two two-way alignments between the sequence of tokens of conflicting regions $a$ (respectively $b$) with respect to that of the original program $o$. For each pair of aligned token sequence we extract an "edit sequence" that represents how to turn the second sequence into the first. These edit sequences – $\Delta_{ao}$ and $\Delta_{bo}$ – are comprised of

the following editing actions (kinds of edits): **=** represents equivalent tokens, **+** represents insertions, **-** represents deletions, $\leftrightarrow$ represents a replacement, and $\emptyset$ is used as a padding token. Overall, this produces four token sequences and two edit sequences: $(a|_o, o|_a, \text{and } \Delta_{ao})$ and $(b|_o, o|_b, \text{and } \Delta_{bo})$. Each token sequence covers the corresponding conflicting region and, potentially, surrounding code tokens. Fig. 2 shows an example of edit sequence.

## 5.2 CONTEXT ENCODING

We pretrain a BERT model $\mathcal{E}$ following the masked language modeling objective on a multilingual dataset of source code files. In each source code file, a set of tokens is sampled at random uniform and replaced with `[MASK]` symbols, and the model aims to reconstruct the original sequence. We make use of a Byte-Pair Encoding (BPE) unsupervised tokenization procedure to avoid a blowup in the vocabulary size given the sparse nature of code identifiers Karampatsis et al. (2020). Besides code tokens, the vocabulary includes the special symbols representing editing steps and the `[MASK]` symbol.

During finetuning, we introduce an edit type embedding combining it with token and position embeddings via addition: $\mathcal{S} = \mathcal{S}_{\mathcal{T}} + \mathcal{S}_{\mathcal{P}} + \mathcal{S}_{\mathcal{E}}$. Edit type embedding helps the model recognize the edit steps, which are not supplied during pretraining. See Fig. 6 in the Appendix for details.

As shown in Fig. 2, we utilize pretrained encoder model to independently encode each of the four token sequences $(a|_o, o|_a, b|_o, \text{and } o|_b)$ of merged programs, passing edit sequences $(\Delta_{ao} \text{ and } \Delta_{bo})$ as type embedding indices.

## 5.3 MERGE TUPLE SUMMARIZATION

In standard sequence learning tasks there is one input and one output sequence. In merge conflict resolution setting, there are multiple input programs and one resolution. To facilitate learning in this setting, we construct MergeBERT as a multi-input encoder neural network, which first encodes token sequences $a|_o, o|_a, b|_o, \text{and } o|_b$, and then aggregates them into a single hidden summarization state:

$$h_m = \sum_{x_i \in (a|_o, o|_a, b|_o, o|_b)} \theta_i \cdot \mathcal{E}(x_i, \Delta) \qquad (3)$$

where $\mathcal{E}$ is the context encoder and $\mathcal{E}(x_i, \Delta)$ are the embedding tensors for each of the sequences $x_i \in (a|_o, o|_a, b|_o, o|_b)$. After encoding and aggregation a linear classification layer with `softmax` is applied:

$$p(r_j | a_j, b_j, o_j) = \text{softmax}(W \cdot h_m + b) \qquad (4)$$

The resulting line-level resolution region is obtained by concatenating the prefix `pref`, predicted token-level resolution $r_j$, and the suffix `suff`. Finally, in the case of a one-to-many correspondence between the original line-level and the token-level conflicts (see 11.5 in the Appendix for an example and 11 for pseudocode), MergeBERT uses a standard beam-search to decode the most promising predictions.

## 6 DATASET

To create a dataset for pretraining, we clone all non-fork repositories with more than 20 stars in GitHub that have C, C++, C#, Python, Java, JavaScript, TypeScript, PHP, Go, and Ruby as their top language. The resulting dataset comprises over 64 million source code files.

The finetuning dataset is mined from over 100 thousand open source software repositories in multiple programming languages with merge conflicts. It contains commits from git histories with exactly two parents, which resulted in a merge conflict. We replay `git merge` on the two parents to see if it generates any conflicts. Otherwise, we ignore the merge from our dataset. We follow Dinella et al. (2021) to extract resolution regions—however, we do not restrict ourselves to conflicts with less than 30 lines only. Lastly, we extract token-level conflicts and labels from line-level conflicts and resolutions. Tab. 4 provides a summary of the finetuning dataset.

# 7 BASELINE MODELS

## 7.1 LANGUAGE MODEL BASELINE

Neural language models (LMs) have shown great performance in natural language generation (Radford et al., 2019; Sellam et al., 2020), and have been successfully applied to the domain of source code (Hindle et al., 2012; Svyatkovskiy et al., 2020; Feng et al., 2020). We consider the generative pretrained transformer language model for code (GPT-C) and appeal to the naturalness of software (Allamanis et al., 2018) to construct our baseline approach for the merge resolution synthesis task. We establish the following baseline: given an unstructured line-level conflict produced by `diff3`, we take the common source code prefix `Pref` acting as user intent for program merge. We attempt to generate an entire resolution region token-by-token using beam search. As an ablation experiment, we repeat this for the conflicts produced with the token-level differencing algorithm (Fig. 1 shows details about prefix and conflicting regions).

## 7.2 DEEPMERGE: NEURAL MODEL FOR INTERLEAVINGS

Next, we consider DeepMerge (Dinella et al., 2021): a sequence-to-sequence model based on the bidirectional GRU summarized in section 3. It learns to generate a resolution region by choosing from line segments present in the input (line interleavings) with a pointer mechanism. We retrain the DeepMerge model on our TypeScript dataset.

## 7.3 JDIME

Looking for a stronger baseline, we consider JDIME, a Java-specific merge tool that automatically tunes the merging process by switching between structured and unstructured merge algorithms (Apel et al., 2012). Structured merge is abstract syntax tree (AST) aware and leverages syntactic information to improve matching precision of conflicting nodes. To compare the accuracy of JDIME to that of MergeBERT, we use the Java test and complete the following evaluation steps: first, we identify the set of merge conflict scenarios where JDIME did not report a merge conflict, and the standard `diff3` algorithm did. Second, we compare the JDIME output to the version of the code where the merge conflict is resolved. Third, we calculateJDIME accuracy by identifying the number of merges where JDIME output file correctly matches the resolved conflict file.

As a result of its AST matching approach, code generated by JDime is reformatted, and the original order of statements is not always preserved. In addition, source code comments that are part of conflicting code chunks are not merged.

A simple syntactic comparison is too restrictive, and JDime merge output can still be semantically correct. To accurately identify semantically equivalent merges, we use GumTree Falleri et al. (2014), an AST differencing tool, to compute fine grained edit scripts between the two merge files. By ignoring semantically equivalent differences computed by GumTree (such as moved method declarations) we have a more accurate baseline comparison between the number of semantically equivalent merges generated by JDime and MergeBERT.

## 7.4 JSFSTMERGE

Apel et al. (2011) introduced FSTMerge, a semi-structured merge engine using an approach similar to JDIME, but that that allows a user to provide an annotated language grammar specification for any language. Trindade Tavares et al. (2019) implemented jsFSTMerge by adapting an off-the-shelf grammar for JavaScript to address shortcomings of FSTMerge and also modifying the FSTMerge algorithm itself. For example, statements can be intermingled with function declarations at the same syntactic level, and statement order must be preserved while function order does not. jsFSTMerge allows for certain types of nodes to maintain their relative order (*e.g.*, statements) while others may be order independent (*e.g.*, function declarations) even if they share the same parent node.

For cases where jsFSTMerge produces a resolution that does not match the user resolution, we manually inspect the output for semantic equivalence (e.g., reordered import statements).

# 8 EVALUATION

We evaluate MergeBERT's accuracy of resolution synthesis. Our evaluation metrics are precision and recall of verbatim string match (modulo whitespaces or indentation) of the decoded predictions to the user resolution extracted from real-world merge resolutions. This definition is rather restrictive as a predicted resolution might differ from the true user resolution by, for instance, only the order of statements, being semantically equivalent otherwise. As such, this evaluation approach gives a lower bound of the MergeBERT model performance.

In addition to the precision and recall, we estimate the fraction of syntactically correct (or parseable) source code suggestions to filter out merge resolutions with syntax errors.

## 8.1 BASELINE MODEL EVALUATIONS

Table 1: Evaluation results for MergeBERT and various neural baselines calculated for merge conflicts in TypeScript programming language test set. The table shows top-1 precision, recall, and F-score metrics.

| Approach | Granularity | Precision | Recall | F-score |
|---|---|---|---|---|
| LM | Line | 3.6 | 3.1 | 3.3 |
| DeepMerge | Line | 55.0 | 35.1 | 45.3 |
| `diff3` | Token | 82.4 | 36.1 | 50.2 |
| LM | Token | 49.7 | 48.1 | 48.9 |
| DeepMerge | Token | 64.5 | 42.7 | 51.4 |
| MergeBERT | Token | 69.1 | 68.2 | 68.7 |

As seen in Tab. 1, performance of language model baselines on the task of merge resolution synthesis is relatively low, suggesting that the naturalness hypothesis is insufficient to capture the developer intent when merging programs. This is perhaps not surprising given the notion of precision that does not tolerate even a single token mismatch.

MergeBERT tool is based on two core components: token-level `diff3` and a multi-input neural transformer model. The token-level differencing algorithm alone gives a high top-1 precision of 82.4%, with a relatively low recall of only 36.1%. Combined with the neural transformer model, the recall is increased to a total of 68.2%. Note, as a deterministic algorithm token-level `diff3` can only provide a single suggestion.

DeepMerge precision of merge resolution synthesis is quite admirable, showing 55.0% top-1 precision. However, it was only able to produce accurate predictions for 63.8% of the test conflicts, failing to generate predictions for merge conflicts which are not representable as a line interleaving. This type of merge conflicts comprises almost a third of the test set, resulting in a recall of 35.1% which significantly lower than MergeBERT, and 50% lower F-score.

**Ablation Study**

We conduct two ablation experiments to carry out a more precise comparison against the DeepMerge baseline.

First, we evaluate DeepMerge model in combination with the token-level `diff3`. As seen in Tab. 1 overall performance improves, resulting in a 13% higher F-score, but still falling short of MergeBERT by 34% F-score (7% lower precision and 60% lower recall). Second, since DeepMerge is not able to produce accurate predictions for merge conflicts which are not representable as a line interleaving, we restrict our test set to this type of merge conflicts only. Overall, MergeBERT yields 70.6% (78.6%) top-1 (top-3) precision compared to 55.0% (59.1%) for DeepMerge, showing that MergeBERT performance on this merge type is comparable to that on the entire test set.

**Semi-structured Program Merge Baselines**

As can be seen from Tab. 2, jsFSTMerge is only able to produce a resolution for 22.8% of conflicts, and the produced resolution is correct only 15.8% of the time. This is in line with the conclusions of the creators of jsFSTMerge that semi-structured merge approaches may not be as advantageous

Table 2: Comparison of MergeBERT to JDime and jsFSTMerge semi-structured merge tools. The table shows top-1 precision, recall, F-score of merge resolution synthesis, fraction of merge conflicts an approach generated resolutions for, and percentage of syntactically correct predictions. JDime evaluation is on a Java data set and jsFSTMerge is on a JavaScript data set.

| Approach | Language | Precision | Recall | F-score | Syntax (%) |
|---|---|---|---|---|---|
| JDime | Java | 26.3 | 21.6 | 23.7 | 90.9 |
| MergeBERT | Java | **63.9** | **63.2** | **63.5** | **98.3** |
| jsFSTMerge | JavaScript | 15.8 | 3.6 | 5.9 | 94.4 |
| MergeBERT | JavaScript | **66.9** | **65.6** | **66.2** | **97.4** |

Table 3: Detailed evaluation results for monolingual and multilingual MergeBERT models, as well as zero-shot performance on an unseen language. The table shows precision, recall, F-score of merge resolution synthesis, and percentage of syntactically correct predictions. Top: monolingual models, bottom: multilingual.

| Test (Train) Languages | Precision | | Recall | | F-score | | Syntax (%) |
|---|---|---|---|---|---|---|---|
| | Top-1 | Top-3 | Top-1 | Top-3 | Top-1 | Top-3 | |
| JavaScript (JS) | 66.9 | 75.4 | 65.6 | 73.9 | 66.2 | 74.6 | 97.4 |
| TypeScript (TS) | 69.1 | 76.6 | 68.2 | 75.6 | 68.7 | 76.1 | 97.0 |
| Java (Java) | 63.9 | 76.1 | 63.2 | 75.2 | 63.5 | 75.6 | 98.3 |
| C# (C#) | 68.7 | 76.4 | 67.3 | 74.8 | 68.0 | 75.6 | 98.3 |
| JavaScript (JS, TS, C#, Java) | 66.6 | 75.2 | 65.3 | 73.8 | 65.9 | 74.5 | 97.4 |
| TypeScript (JS, TS, C#, Java) | 68.5 | 76.8 | 67.6 | 75.8 | 68.0 | 76.3 | 96.9 |
| Java (JS, TS, C#, Java) | 63.6 | 76.0 | 62.9 | 75.1 | 63.2 | 75.6 | 98.2 |
| C# (JS, TS, C#, Java) | 66.3 | 76.2 | 65.1 | 74.8 | 65.7 | 75.5 | 98.3 |
| Scala (JS, TS, C#, Java) | 57.8 | 64.8 | 56.5 | 63.3 | 57.1 | 64.1 | 97.9 |

for dynamic scripting languages (Trindade Tavares et al., 2019). Because jsFSTMerge may produce reformatted code, we manually examined cases where a resolution was produced but did not match the user resolution (our oracle). If the produced resolution was semantically equivalent to the user resolution, we classified it as correct.

Tab. 3 shows the detailed evaluation results of the MergeBERT. As seen, multilingual variant of MergeBERT yields $63.6 - 68.5\%$ top-1 and $75.2 - 76.8\%$ top-3 precision of verbatim match and relatively high recall values. Overall, the multilingual variant of the model generates results comparable to the monolingual versions on the languages present in the training set and shows the potential for zero-shot generalization to unseen languages. We test the zero-shot generalization property on merge conflicts in Scala[1] programming language and obtain an encouraging 64.8% top-3 precision of merge resolution synthesis.

## 9 CONCLUSION

This paper introduces MergeBERT, a transformer-based program merge framework that leverages token-level differencing and reformulates the task of generating the resolution sequence as a classification task over a set of primitive merge patterns extracted from real-world merge commit data. MergeBERT exploits pretraining over massive amounts of code and then finetuning on specific programming languages, achieving 64–69% precision and 63–68% recall of merge resolution synthesis. Lastly, MergeBERT is flexible and effective, capable of resolving more conflicts than the existing tools in multiple programming languages.

Our work focuses on helping software developers resolve merge conflicts and improve their productivity. The lightweight finetuning approach that lies at the core of this tool promotes the re-usability of pretrained transformer models for software engineering tasks.

---

[1] https://www.scala-lang.org/

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

# 10 APPENDIX

## 10.1 PRIMITIVE MERGE RESOLUTION TYPES

Given a merge tuple $(\mathcal{A}, \mathcal{B}, \mathcal{O}, \mathcal{M})$ with line-level conflicting regions $A_i, B_i, O_i$, i=0...N, and token-level conflicting regions $a_j^i, b_j^i, o_j^i$ corresponding to a line-level conflict $i$, we define following nine basic merge resolution types which serve as labels for supervised classification task:

1. Take changes $a_j^i$ proposed in program $\mathcal{A}$ (developer branch A) as resolution,

2. Take changes $b_j^i$ proposed in program $\mathcal{B}$ as resolution,

3. Take changes $o_j^i$ in the base reference program $\mathcal{O}$ as resolution,

4. Take a string concatenation of changes in $a_j^i$ and $b_j^i$ as resolution,

5. Take a string concatenation of changes in $b_j^i$ and $a_j^i$ as resolution (reverse order as compared to the previous),

6. Take changes $a_j^i$ proposed in program $\mathcal{A}$, excluding the lines also present in the base reference program $\mathcal{O}$, as resolution,

7. Take changes $b_j^i$ proposed in program $\mathcal{B}$, excluding the lines present in the base, as resolution,

8. Take a string concatenation of changes in $a_j^i$ and $b_j^i$, excluding the lines present in the base, as resolution,

9. Take a string concatenation of changes in $b_j^i$ and $a_j^i$, excluding the lines present in the base, as resolution (reverse order as compared to the previous),

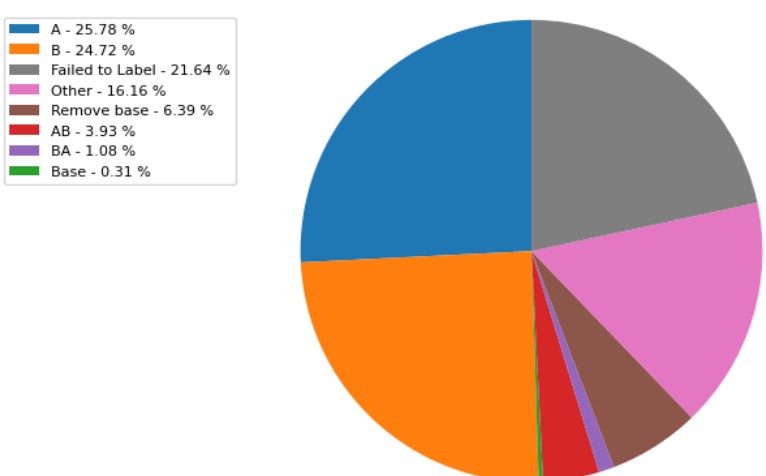

Figure 3: Summary of merge conflict resolution labels in our dataset for TypeScript: label distribution for merge conflicts extracted with the standard (line-level) diff3 algorithm, right.

While the above 9 primitive merge patterns are defined for token-level conflicting chunks, same definition could be applied at line-level, substituting $a^i_j, b^i_j, o^i_j$ chunks with $A_i, B_i, O_i$.

We use a data-driven approach to identify these 9 primitive merge resolution patterns based on the analysis of the real-world merge conflict resolutions from GitHub. Our analysis shows that over 99% of all the merge conflicts can be represented using these labels. While the above nine resolution types are primitive, they form a basis sufficient to cover a large class of real-world merge resolutions in modern version control systems, including arbitrary combinations or interleavings of lines.

Fig. 3 shows the label distribution obtained for the standard (line-level) diff3 conflicting regions in our dataset for TypeScript programming language. As seen, nearly 50% of all cases are trivial – take changes from branch A or B. Arguably, these cases can be resolved without machine learning and are easily addressed by *take ours* or *take theirs* merge resolution strategies. The "Remove base" category combines the four primitive merge patterns that remove the lines present in the base branch from the resolution (e.g. take changes proposed in one program and exclude the lines present in base). The "Other" category consists of arbitrary line-interleavings.

Fig. 4 shows the label distribution obtained for token-level differencing algorithm. It excludes trivial (take A or take B) merge resolutions at line-level. Note, that "take A" merge resolution at token-level does not correspond to "take ours" or "take theirs" merge resolution strategy, and can map to any label at line-level, thus representing a non-trivial merge scenario stimulating for machine learning studies.

It is important to stress, these primitive merge resolution types at token-level are not strictly defined *templates* dictating which syntactic structures should be selected from input programs. For instance, a label "take changes proposed in program $\mathcal{A}$" can correspond to a single code token as well as an entire method signature or body. As such, the merge types are not restrictive in their representation power of merge conflicts.

## 10.2   SUMMARY OF THE FINETUNING DATASET

Tab. 4 shows the summary of the finetuning merge conflict resolution dataset. The finetuning dataset is split into development and test set in the proportion 80/20 at random at file-level. The development set is further split into train and validation set in 80/20 proportion at the merge conflict level.

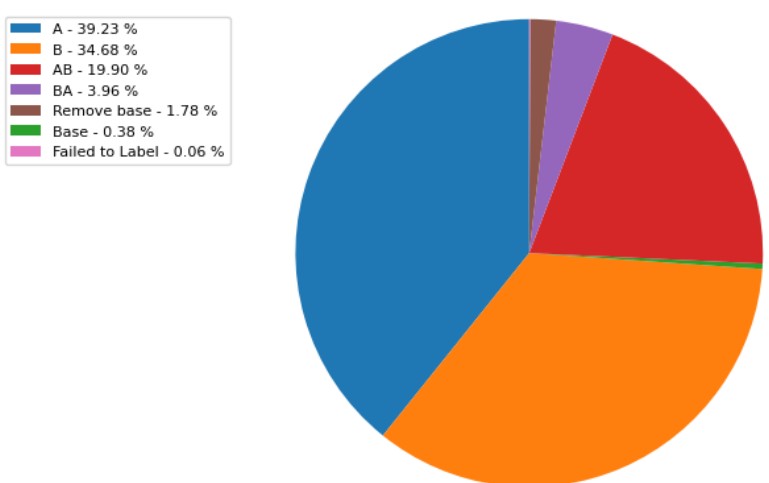

Figure 4: Summary of merge conflict resolution labels in our dataset for TypeScript: label distribution for merge conflicts extracted with token-level differencing algorithm.

Table 4: Number of merge conflicts in the dataset.

| Programming language | Development set | Test set |
|---|---|---|
| C# | 27874 | 6969 |
| JavaScript | 66573 | 16644 |
| TypeScript | 22422 | 5606 |
| Java | 103065 | 25767 |
| Scala | | 389 |

## 11 MERGE RESOLUTION DECODING

Each model prediction yields a probability distribution $p(r_j|a_j, b_j, o_j)$ over primitive merge patterns given a token-level conflict. In case of a one-to-many correspondence between original line-level and the token-level conflicts (see, for instance, Fig. 8) to approximate the original $p(R|A, B, O)$ we decode the most promising combination from the predicted solution space. This can be conceptualized as a maximum cost path search on a matrix, which we approach via a beam search algorithm.

---

**Algorithm 1** Merge conflict resolution decoding algorithm (beam search) with MergeBERT

---
$b \leftarrow \{r_0, 0\}$        ▷ Initialize beam {state, logprob}
$\{a_j, b_j, o_j, r_j\}_{j=0...N} \leftarrow \texttt{token\_diff3}(\mathcal{A}, \mathcal{B}, \mathcal{O}, \mathcal{M})$,    ▷ Perform token-level differencing
**for** $j \leftarrow 0$ **to** $N$ **do**
    **if** $a_j = \emptyset \wedge b_j = \emptyset \wedge o_j = \emptyset$ **then**     ▷ In case token-level merge results in a clean merge
      **return** $pref_j :: r_j :: suff_j$
    **else**
      **for** $(r_j^k, p_j^k) \in \textbf{TopK}(\texttt{mergebert}(r_j|a_j, b_j, o_j))$ **do**
        $b \leftarrow b \cup \{r + pref_j :: r_j^k :: suff_j, p + p_j^k\}$    ▷ Update beam for each token-level
conflict $j$
      **end for**
      $b \leftarrow \textbf{TopM}(b)$        ▷ Prune candidates to keep top-M
    **end if**
**end for**
$R \leftarrow b[0]$        ▷ Get resolution region string
**return** $R$

---

As a result, the model prediction for each line-level conflict consists of either a label for a token-level conflict or a combination of labels for multiple token-level conflicts representing the best prediction for each token-level conflict within the line-level conflict. Given these labels for each line-level conflict and the contents of the merged file, MergeBERT generates the code corresponding to the resolution region. The contents of the merged file includes the conflict in question and its surrounding regions. Therefore, for each conflicting line, MergeBERT chooses between the versions of code based on the labels the model produced and generates the resolution code by concatenating them. Afterwards, MergeBERT checks the syntax of the generated resolved code, and in case of correctness, outputs it as the candidate merge conflict resolution.

In case of multiple line-level conflicts in the merged file, MergeBERT refines the contents of the merged file that serves as the surrounding region of the conflict. More specifically, for each line-level conflict, MergeBERT replaces the other conflicts in the the merged file contents with the code it previously generated as their predicted resolutions. For this purpose, MergeBERT updates the contents of the merged file after resolving each line-level conflict with the code it generates as the conflict resolution based on the model prediction.

## 11.1 INFERENCE COST

Computational efficiency is an important constraint influencing machine learning design decisions in production environments (e.g. deployment in IDE, GitHub action). In the following, we discuss inference costs and floating point operations per second (FLOPs) of MergeBERT as compared to the language model baseline.

In this paper, we reformulate the task of merge conflict region as a classification problem. This provides a major speedup during inference, due to a smaller number of inference calls necessary to decode a resolution. Indeed, in most cases MergeBERT requires only 1 inference call to resolve a merge conflict, with up to 3 calls in the worst case, based on our dataset. The cost of a single inference call on a 16GB Tesla V100 GPU is 60 ms. The end-to-end time to resolve a merge conflict (including tokenization, alignment, and edit sequence extraction) is 105 ms on average, and up to 500 ms in the worst case.

With GPT-C language model, the resolution region is decoded token-by-token via the beam search algorithm. The average time to decode a single token (in our experiments we use beam width of 5, and 1024 tokens context length, with past hidden state caching optimization enabled) on a 16GB Tesla V100 GPU is about 15 ms. With token-level differencing, the resolution size is 70 tokens on average (up to 1584 tokens maximum, in our dataset), which yields 1.1 seconds on average and up to 23.8 seconds in the worst case (the largest conflict) to generate resolution token sequence. Overall, end-to-end inference time required to resolve a merge conflict (including parsing and tokenization) is 2.3 seconds on average and up to 48.5 seconds for the largest conflict. From the user experience prospective in IDE, inference times of over 10 seconds are prohibitively slow.

### 11.1.1 FLOATING POINT OPERATIONS PER SECOND

In the following, we identify main operations in the transformer encoder, for the multi-input Merge-BERT architecture (see Fig. 2 for reference):

- Self-attention: 600 MFLOPs x 4 inputs (encoder weights are shared for all inputs),
- Feed-forward layer: 1200 MFLOPs x 4 inputs.

Contribution of the lightweight pooling (aggregation) and classification layers are negligibly small. With a total of 6 transformer encoder layers this yields: 43200 MFLOPs per forward pass.

For the GPT-C transformer decoder-only model we get:

- Self-attention: 600 MFLOPs
- Feed-forward layer: 1200 MFLOPs

with a total of 12 encoder layers this yields: 21600 MFLOPs per inference call, and for 6 encoder layers: 10800 MFLOPs.

With larger FLOPs per a single forward pass as compared to generative approach, with MergeBERT we gain a significant reduction in total FLOPS required to decode resolution region as a result of needing to performing orders of magnitude less calls (1–3 calls with MergeBERT as compared to 70–1584 with a language model), making this approach an appealing candidate for deployment in IDE.

## 11.2 IMPACT OF PRETRAINING DETAILS

The effect of self-supervised pretraining is two-fold: (1) it speeds up the time to solution as a result of faster model convergence – we observe a 20% higher F-score after 5 training epochs – and 32 times larger finetuning training throughput, and (2) it yields 14% overall higher F-score as compared to a model trained from scratch.

While experiments with various pretraining objectives (Guo et al., 2021; Rozière et al., 2021) is outside of the scope of this work, as a baseline, we utilized the CodeBERT public checkpoint for a downstream task of merge conflict resolution which utilizes a similar pretraining objective as ours. The resulting model showed a comparable F-score, and a likely explanation for the difference is that the CodeBERT is pretrained on the CodeSearchNet dataset, which does not include C# and TypeScript programming languages used in this study.

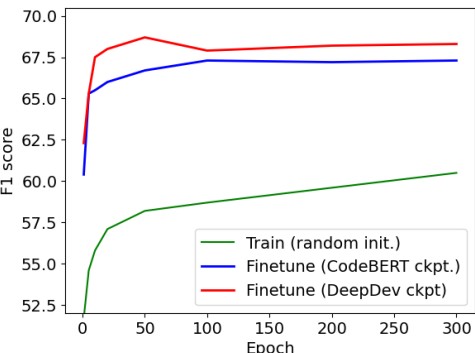

Figure 5: MergeBERT model trained from scratch as compared to finetuning training for sequence classification downstream task with the encoder weights transferred and frozen during finetuning. The F-scores of merge resolution synthesis are quoted for TypeScript test set as a function of epoch. Finetuning performance with CodeBERT-base[2] publicly available checkpoint is quoted for reference.

## 11.3 CONTEXT ENCODING DETAILS

Fig. 6 illustrates how input representations are extracted.

We conduct an ablation study on the edit type embedding to understand the impact of edit-awareness of encoding on the model performance. As shown in Tab. 5, edit type embedding information could improve MergeBERT F-score by 7%.

Table 5: Evaluation results for MergeBERT and the model variant without edit-type embedding for merge conflicts in TypeScript programming language test set. The table shows top-1 precision, recall, and F-score metrics.

| Approach | Precision | Recall | F-score |
|---|---|---|---|
| w/o type embedding | 65.2 | 63.1 | 64.1 |
| MergeBERT | **69.1** | **68.2** | **68.7** |

---

[2]https://huggingface.co/microsoft/codebert-base

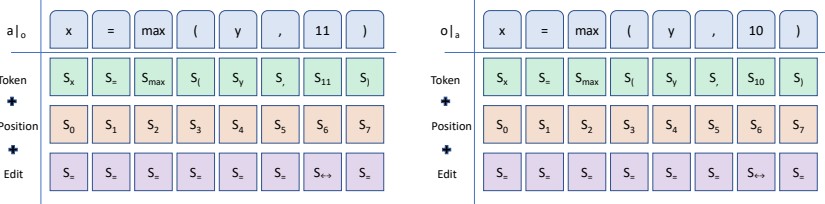

Figure 6: MergeBERT input representation. The input embeddings are a sum of the token embeddings, the type embeddings and the position embeddings. Type embeddings are extracted from the edit sequence step that represent how to turn the $a|_o$ sequence into the $o|_a$

### 11.4 Implementation Details

We pretrain a BERT model with 6 encoder layers, 12 attention heads, and a hidden state size of 768. The vocabulary is constructed using byte-pair encoding method (Sennrich et al., 2016) and the vocabulary size is 50000. We set the maximum sequence length to 512. Input sequences cover conflicting regions and surrounding code (i.e., fragments of `Pref` and `Suff`) up to a maximum length of 512 BPE tokens. The backbone of our implementation is HuggingFace's `RobertaModel` and `RobertaForSequenceClassification` classes in PyTorch, which are modified to turn the model into a multi-input architecture shown in Fig. 2.

We train the model with Adam stochastic optimizer with weight decay fix using a learning rate of 5e-5, 512 batch size and 8 backward passes per `allreduce` on 64M files in C, C++, C#, Python, Java, JavaScript, TypeScript, PHP, Go and Ruby programming languages. The training was performed on 64 NVIDIA Tesla V100 with 32GB memory for 21 days; we utilized mixed precision. Finetuning was performed on 4 NVIDIA Tesla V100 GPUs with 16GB memory for 6 hours.

In the inference phase, the model prediction for each line-level conflict consists of one or more token-level predictions. Given the token-level predictions and the contents of the merged file, MergeBERT generates the code corresponding to the resolution region. The contents of the merged file include the conflict in question and its surrounding regions. Afterward, MergeBERT checks the syntax of the generated code with tree-sitter[3] parser and outputs it as the candidate merge conflict resolution only in case of correctness.

The model prediction for each line-level conflict consists of either a label for a token-level conflict or a combination of labels for multiple token-level conflicts representing the best prediction for each token-level conflict within the line-level conflict. Given these labels for each line-level conflict and the contents of the merged file, MergeBERT generates the code corresponding to the resolution region. The contents of the merged file includes the conflict in question and its surrounding regions. Therefore, MergeBERT, for each conflicting line, chooses between the versions of code based on the labels the model produced and generates the resolution code by concatenating them. Afterwards, MergeBERT checks the syntax of the generated resolved code, and in case of correctness, outputs it as the candidate merge conflict resolution.

In case of multiple line-level conflicts in the merged file, MergeBERT refines the contents of the merged file that serves as the surrounding region of the conflict. More specifically, MergeBERT replaces the other conflicts in the the merged file contents with the code it previously generated as their predicted resolutions. For this purpose, as shown in Fig. 7, MergeBERT updates the contents of the merged file after resolving each line-level conflict with the code it generates as the conflict resolution based on the model prediction.

### 11.5 Multiple Conflicting Regions

MergeBERT can deal with non-trivial real-world merges, composed of multiple conflicting chunks. To provide an example of such a merge conflict, we include Fig. 8. MergeBERT correctly predicts

---

[3]https://tree-sitter.github.io/tree-sitter/

Prefix for conflict 1

```
...
some code: 1
...
// [CONFLICT 1] – start
<<<<<<< Current Change (A)
    code of conflict 1 version A
||||||| Base
    code of conflict 1 version
Base
=======
    code of conflict 1 version B
>>>>>>> Incoming Change (B)
// [CONFLICT 1] – end
...
some code: 2
...
// [CONFLICT 2] – start
<<<<<<< Current Change (A)
    code of conflict 2 version A
||||||| Base
    code of conflict 2 version
Base
=======
    code of conflict 2 version B
>>>>>>> Incoming Change (B) //
[CONFLICT 2] – end
...
```

(a) Resolving conflict 1

Prefix for conflict 2

```
...
some code: 1
...
code of predicted resolution
...
some code: 2
...
// [CONFLICT 2] – start
<<<<<<< Current Change (A)
    code for conflict 2 version A
||||||| Base
    code for conflict 2 version
Base
=======
    code for conflict 2 version B
>>>>>>> Incoming Change (B) //
[CONFLICT 2] – end
...
```

(b) Resolving conflict 2

Figure 7: An example of a file with multiple conflicting regions.

a concatenation of changes proposed by developers A and B for the first token-level chunk and a concatenation of changes proposed by developers B and A (in the reverse order) for the second chunk.

```
Var form = $(this);
settings.dom.bind(settings.event, function() {
    var status = false;
    var data = form.serialize();
    if(settings.fields){
<<<<<<< A.js
        data += '&' + $.param({'fields':settings.fields});
||||||| Base.js
        params.fields = settings.fields;
=======
        data += $.param({fields:settings.fields});
>>>>>>> B.js
    }
    if (status && settings.submitHandler){
        return settings.submitHandler.apply(this);
    }
Return status;
});
```

(a) Line-level conflict

```
var form = $(this);
settings.dom.bind(settings.event, function()  {
    var status = false;
    var data = form.serialize();
    if (settings.fields) {
        data += '
<<<<<<< A.js
        data += '&' + $
||||||| Base.js
        params
=======
        data += $
>>>>>>> B.js
        .
<<<<<<< A.js
        param({'fields':
||||||| Base.js
        fields =
=======
        param({fields:
>>>>>>> B.js
settings.fields});
    if (status && settings.submitHandler) {
        return settings.submitHandler.apply(this);
    }
    return status;
});
```

(b) Token-level conflicts

```
Var form = $(this);
settings.dom.bind(settings.event, function() {
    var status = false;
    var data = form.serialize();
    if(settings.fields){
        data += $.param({'fields':settings.fields});
    }
    if (status && settings.submitHandler){
        return settings.submitHandler.apply(this);
    }
Return status;
});
```

(c) Resolved merge

Figure 8: Example real-world merge conflict resolved by MergeBERT. (Top) merge conflict represented through the standard `diff3`, (middle) corresponding token-level conflicts, and (bottom) the user resolution.

