# OpenReview forum: "MergeBERT: Program Merge Conflict Resolution via Neural Transformers"
_ICLR.cc/2022/Conference — ICLR 2022 Submitted_

### Official Review · Reviewer_TUUK · 2021-11-02

**Correctness:** 3
**Technical Novelty And Significance:** 2
**Empirical Novelty And Significance:** 2
**Recommendation:** 3
**Confidence:** 4

**Main Review:**

Pros:
+ This is a compelling task, merge conflicts are an important problem
+ Reducing the problem to using a classifier is valuable practically
+ The nine primitive merge resolution patterns covering 99% of merge cases is an interesting observation

Cons:
+ The encoding is token-based and rather standard
+ Missing ablation study
+ Experimental evaluation partial with respect to most important baseline

* Tackling the merge conflicts problem is interesting. Using a learned model to decide how to resolve it is interesting as well (but not novel), since, it’s not always clear what is the best resolution (which is consistent with the context).

* It’s not clear how the model looks like, Do you train BERT only for word vectors? If so, what about exploiting the context? Is the encoder of the aligned sequences the pre-trained BERT? It looks like you are not finetuning it (based on the blue squares of Figure 2, showing that only the Edit Type Embeddings are finetuned), so how should it handle the new kind of inputs? (token embedding + type embedding + position embeddings)

* How important is the edit type embedding used in finetuning? The paper refers to Figure 5 in the appendix that shows how these are used, but not how important are they for the final result. In general, the paper would benefit from a more extensive ablation study.

* The comparison with DeepMerge includes cases where DeepMerge cannot produce a prediction, as these cases are not line-level merges. This is not an apples-to-apples comparison. What would happen if we compare DeepMerge/MergeBert considering only line-level merges?

* Comparisons to JDime and jsFSTMerge are useful for showing the superiority of your neural approach, but I am more interested in understanding what makes your approach work well.

* Table 3 is pretty much due to properties of BERT and not of MergeBERT, right? Similar experience reported in CodeBERT?

* Primitive Merge Resolution Types: Appendix 10.1 seems to emphasize this, but I do have some comments:
"Our analysis shows that over 99% of all the merge conflicts can be represented using these labels"
But FIgure 3 (left) says that you failed to label 21.64% of the TypeScript examples. Can you please elaborate on this point?
It’s not clear how the colors in Figure 3 are related to your 9 classes. Does “Remove base” correspond to classes 6-9?, is so, what does pink (“Other”) refer to?
Please align the legends on both sides of Figure 3.
I would emphasize that there are almost 21.64% non-trivial examples (in TypeScript), which the token-level diff method can solve by using a classification approach.

* It would be helpful to add an encoder-decoder baseline - given (A, B, O), generate the resolution.

* Table 1- what is the meaning of the diff3 row? I guess it’s for the examples where the token granularity diff managed to solve conflicts the line-diff couldn’t. Please clarify that.

* Table 1 - what about using BERTMerge with line diff and DeepMerge with token diff? It’s essential for a fair comparison.

* Section 8.2: The impact of pretraining is not disputed, but again it is not a contribution of the technique presented here.

Minor:

* Figure 2: should all edit operations in the blue squares be <->?

* Figure 7(b) needs to be corrected: `data +=` is also a conflict, and there is a piece of code after the last conflict - ``});`.


**Summary Of The Paper:**

This paper is about using BERT for the automatic resolution of merge conflicts. The main idea is to cast the problem of automatic merge resolution as a classification problem with 9 classes (token-level merge patterns). The technique use token-level differencing to represent the input to the merge problem as four aligned token sequences, together with two edit-type embeddings. The input sequences are fed to BERT and the results are aggregated and used for classification.
The technique is compared to state-of-the-art automatic-merge tools and is shown to outperform them in all benchmarks.


**Summary Of The Review:**

The problem of automatic merge resolution is compelling. The authors present a solution that is based on token-level diff (no ML contribution), identify 9 merge patterns (no ML contribution), and finally use BERT with minor adaptations to classify which merge pattern should be used. For this model, some details of what exactly is being fine-tuned are not clear.

The suggested approach shows improvement compared to existing baselines.

The experimental evaluation is hard to follow because neither of the existing baselines addresses the exact same kind of merges. The closest baseline in terms of technique is DeepMerge, and the comparison to that baseline is quite partial.

Overall, this may be a great software-engineering contribution, but I'm afraid that there's not a lot here for ICLR.

---

> ### Author Response · Authors · 2021-11-13
> **Responses to reviewer TUUK**
>
> Thank you for engaging with our submission.
>
> **Re: Overall, this may be a great software-engineering contribution, but I'm afraid that there's not a lot here for ICLR.**
> A: We respectfully disagree that there is not a lot for ICLR.  Understanding, modeling, and generating code is of huge interest to ICLR.  The “merge resolution” problem studied in our paper is a great task to elucidate different modeling techniques for code generation.  The dataset is plentiful (merges are common and easily mined from Github), the solutions are simpler than a traditional code generation task as most resolutions do not require generating new tokens, and finally, as the reviewer notes, “solving” this problem would have a positive impact in the daily lives of a huge number of people.  We hope this is the first of many papers on this topic in ICLR.
>
> **Q: Experimental evaluation partial with respect to most important baseline. What about using BERTMerge with line diff and DeepMerge with token diff? It’s essential for a fair comparison.**
> The reviewer raises several related issues regarding the baselines, and we address them all here.
>
> Following is the DeepMerge performance with token diff (also included in Table 1 of new version):
>
> | Approach | Precision | Recall | F-score |
> |-------|-------|-------|-------|
> |DeepMerge  | 64.5 | 42.7 | 51.4 |
> | MergeBERT  | 69.1 | 68.2 | 68.7 |
>
> The overall DeepMerge performance improves, resulting in a 13\% higher F-score, but still falling short of MergeBERT by 34\% F-score (7\% lower precision and 60\% lower recall).
>
> FIgure 3 (left) shows that over 20% of all merges require token-level interleaving. Thus, any line-based approach cannot address this large subset of merges (including DeepMerge). A line-level version of MergeBERT’s recall would drop by at least this much and is also motivation for why MergeBERT’s approach is superior to DeepMerge. In contrast, DeepMerge trained at the token-level granularity is an interesting area to pursue but is not a simple extension of the prior work. When investigating the data, we found almost all token-level merges fell into one of our 9 classes (See Figure 3), and thus a classification-based approach does not give up on recall. The architecture of MergeBERT is designed to exploit this fact. Thus, we respectfully disagree that adding MergeBERT trained on line-level conflicts (clearly has limitations by only operating on lines) or DeepMerge trained on token-level conflicts (a paper in of itself) is a requirement to show that MergeBERT is a contribution over prior art. We can do a better job explaining this in the paper and will update the text to reflect this discussion.
>
> Following are the results for TypeScript MergeBERT restricted to the line-interleaving merge type only:
>
> |	| Top-1 | Top-3 |
> |-----|----|----|
> | Precision | 70.6 | 79.0 |
> | Recall | 69.9 | 78.3 |
> | F-score | 70.2 | 78.6 |
>
> Comparing to DeepMerge's top-1 precision of 55.0% and top-3 precision of 59.1%.
>
> Lastly, the DeepMerge paper demonstrated a Fairseq based encoder-decoder baseline (on the concatenation of A, B, and O) performed poorly, and was only able to correctly generate resolutions for ~3% of the cases. We can replicate this work for a final version.
>
> **Q: Do you train BERT only for word vectors? If so, what about exploiting the context?**
> A: Can you please clarify what exactly do you refer to as “context”?
> Input sequences to MergeBERT cover token-level conflicting regions and surrounding code (i.e., a prefix and suffix surrounding the token-level conflict region). In comparison, DeepMerge only considers code within a line-level conflict as “context”.
>
> We treat the source code data as a sequence of tokens corresponding to the output of a lexical analyzer. In this work, we do not leverage non-terminal nodes of syntax trees, or global file-level context. This is based on an assumption that most of the information needed for developer to resolve a merge conflict is around the conflicting region itself.
>
> **Q: Is the encoder of the aligned sequences the pre-trained BERT?**
> A: All token sequences (4 total, aligned pairwise) are encoded by BERTs. The edit sequences, however, are not encoded by BERT, but rather are passed as embedding layer. This is done because BERT is not pretrained on edit types.
>
> **Re: finetuning**
> A: Lightweight finetuning has been getting a lot of attention in practice. In this work, we try to promote the re-usability of pretrained transformer models for software engineering tasks. For that reason, we kept most of the parameters frozen and only finetuned type embeddings, aggregation, and output classification layers. Nevertheless, experiments around gradual unfreezing or finetuning end-to-end could yield further improvement, but would be computationally more expensive (we will mention that in the discussion).

---

> > ### Author Response · Authors · 2021-11-16
> > **Responses to reviewer TUUK (part 2)**
> >
> > **Q: How important is the edit type embedding used in finetuning?**
> >
> > A: We added an ablation study on the edit type embedding that demonstrates the impact of edit-awareness of encoding on the model performance. As shown in below, edit type embedding information could improve MergeBERT's F-score by 7\%.
> >
> > | Approach |  Precision | Recall | F-score  |
> > |-----|-----|-----|-----|
> > | w/o type embedding  | 65.2 | 63.1 | 64.1  |
> > | MergeBERT | 69.1 | 68.2 | 68.7 |
> >
> > **Q: how should it handle the new kind of inputs?**
> > A: The underlying BERT model is pretrained on a massive multilingual source code dataset, and the downstream fintuning task of merge conflict resolution is also in the domain of source code. Our BPE vocabulary size is 50000 which is sufficiently large to compose unseen code identifiers not present in the training set. For a more specific answer, we would ask what do you mean by new kinds of inputs (e.g. an unseen language)
> >
> > **Q: Table 3 is pretty much due to properties of BERT and not of MergeBERT, right? Similar experience reported in CodeBERT?**
> > A: Table3 results should be attributed to both the BERT and MergeBERT.
> > BERT is a single input single output model. First, it has been shown in the DeepMerge paper that feeding a concatenation of (A,B,O) as a single input leads to poor performance. Second, we have observed that concatenating the embeddings of (A,B,O) and feeding to pre-trained BERT as single input gives results similar to a randomly initialized model. This is why we proposed MergeBERT architecture, which independently encodes the aligned sequences then aggregate them. Finally, edit-awareness aspect of the model is also crucial.
> >
> > **Q: Primitive Merge Resolution Types: Appendix 10.1 seems to emphasize this, but I do have some comments: "Our analysis shows that over 99% of all the merge conflicts can be represented using these labels" But FIgure 3 (left) says that you failed to label 21.64% of the TypeScript examples. Can you please elaborate on this point?**
> >
> > A: We will made the Appendix 10.1 and the Figure3 more clear in the new version of the paper.
> > The Figure3 left shows the distribution of merge conflicts extracted with the standard (line-level) diff3 algorithm, while the right plot refers to token-level diff3. The corresponding labels do not refer to the same exact cases. For instance, the label “A” in left plot simply corresponds to a merge resolution strategy like “take ours” or “take theirs” and is conceptually different from label “A” on the right (which may correspond to a more complex merge cases, including an interleaving). As plot label shows, we failed to assign a label to 21% of line-level conflicts, but not the token-level cases.
> >
> > **Q: It’s not clear how the colors in Figure 3 are related to your 9 classes. Does “Remove base” correspond to classes 6-9?, is so, what does pink (“Other”) refer to?
> > A: We will clarify this in the caption and improve quality of the plot.
> > The "Remove base" category combines the four primitive merge patterns that remove the lines present in the base branch from the resolution (e.g. take changes proposed in one program and exclude the lines present in base). The "Other" category consists of arbitrary line-interleavings, at the token level (right plot) these merge conflicts are mapped to one of the 9 primitive merge patterns.
> >
> > **Q: I would emphasize that there are almost 21.64% non-trivial examples (in TypeScript), which the token-level diff method can solve by using a classification approach.**
> > A: This statement would be incorrect: Token-level diff is a deterministic algorithm, which does not perform classification. The Figure.3 only shows the statistics about labels assigned to conflicts, which serve as input for supervised learning (finetuning).
> > The unlabeled 21.64\% of conflicts at the line-level may map to various categories at the token-level, and are not the only cases the MergeBERT solves.
> >
> > **Q: Figure 2: should all edit operations in the blue squares be `<->`?**
> > A: The blue squares represent the edit sequence steps for pairwise alignment between Base and B. Since most of those tokens are unchanged, the edit steps should indeed be `==` (`<>` means a swap), but for two inserted tokens `,` and `z` the edit step should be `+`, which are shown incorrectly. Thanks for pointing this out, we have corrected the figure.
> >
> > **Q: Figure 7(b) needs to be corrected: `data +=` is also a conflict, and there is a piece of code after the last conflict - ``});`.**
> > A: Thanks for pointing it out, this is a typo. We have corrected the figure.

---

### Official Review · Reviewer_k5J6 · 2021-11-02

**Correctness:** 3
**Technical Novelty And Significance:** 3
**Empirical Novelty And Significance:** 3
**Recommendation:** 6
**Confidence:** 4

**Main Review:**

Pros:

+ Hierarchical two-level differencing appears to be an intuitive and good idea.
+ Casting the problem as classification over a set of merge patterns appears advantageous in both learning and computation, since most of the time the resolution comes from either of the two change versions.
+ Pre-training can be applied directly.

Cons:

- This idea is somewhat incremental, largely based on the work of Dinella et al., 2020.
- MergeBERT is not as good as *diff3* in terms of precision, which is to me a more important measure.
- Some parts of the techniques and experiments are not clear, for example the class label and data split. I have more detailed questions below.


Questions:

1. It is somewhat unintuitive that `let` is not part of the token-level conflict. I understand that `var` is not changed in the version B, but it is important from both A and B perspectives that there is a change that needs an agreement in resolution. In that case, I don’t think merging `let` right away is an indisputable decision.

2. For this change from `--num_cores=2` to `--max_length=256 --num_cores=2`, what is the result of the two-level diff?

3. Is treating conflicting regions independently in Eq. (2) is oversimplified?

4. How do you split the train and test dataset (Table 4)? Is there a validation dataset?

5. How do you deal with conflicts from more than 2 versions?



**Summary Of The Paper:**

This paper presents MergeBERT, a deep model for resolving program merge conflicts in software development. The authors introduce a new, hierarchical differencing and cast the problem as classifying over a fixed set of merge patterns, instead of generating . The model is pre-trained on a large corpus of GitHub code.

**Summary Of The Review:**

Overall, the paper presents several new technical contributions and insights, but some parts are not entirely convincing.

---

> ### Author Response · Authors · 2021-11-13
> **Response to reviewer k5J6**
>
> Thank you for your kind review and detailed questions.
>
> **Re: MergeBERT is not as good as diff3 in terms of precision**
> A: Besides precision, the recall metric is crucial for real-world scenarios (a model that is correct 1 out of 1000 merges is not very interesting).  MergeBERT has significantly better recall as it is invoked when diff3 fails.
>
> **Q: It is somewhat unintuitive that `let` is not part of the token-level conflict. I understand that `var` is not changed in the version B, but it is important from both A and B perspectives that there is a change that needs an agreement in resolution. In that case, I don’t think merging `let` right away is an indisputable decision.**
> A: This is a good point; however, it can also be applied to the existing git `diff3` algorithm that works on lines. The idea of `diff3` is that if one person changes a line, and the other does not, there is no conflict. We embody this same intuition at the token-level (but of course add a neural component to suggest a resolution when a conflict occurs).
>
> Put another way, suppose we forced B’s edits to happen after A’s edits.  B would only see A’s change to use `let`.  Likewise, suppose B’s edits happen before A’s edits: B would never see A’s change.  In both cases, B does not have the chance to influence A.  Of course, B can later revert A’s edit if they want `var` instead of `let`!
>
>
> **Q: For this change from `--num_cores=2` to `--max_length=256 --num_cores=2`, what is the result of the two-level diff?**
> A: The answer would generally depend on the granularity of diff, and programming language grammar since we use tree-sitter lexical tokenizer to split into tokens. In case of the aligned diff, it would also depend on the order of sequences compared.
> We use the definition of aligned sequence diff from Yin et. al. (“Learning to Represent Edits”, 2019) paper.
> Let’s denote `A = --num_cores=2` and `B = --max_length=256 --num_cores=2`. Using notations of Figure2, we would get:
> 1. diff(A, B)
>
> `A|B`: `['<pad>', '<pad>', '<pad>', '<pad>', '--', 'num_cores', '=', '2']`
>
> `Delta(A, B)`: `[+, +, +, +, =, =, =, =]`
>
> `B|A`: `['--', 'max_length', '=', '256', '--', 'num_cores', '=', '2']`
>
> 2. diff(B, A)
>
> `B|A`: `['--', 'max_length', '=', '256', '--', 'num_cores', '=', '2']`
>
> `Delta(B,A)`: `[-, -, -, -, =, =, =, =],`
>
> `A|B`: `['<pad>', '<pad>', '<pad>', '<pad>', '--', 'num_cores', '=', '2']`
>
> **Q: Is treating conflicting regions independently in Eq. (2) is oversimplified?**
> A: Treating token-level conflicts is a simplifying assumption. However, our data shows that only 5% of merge conflicts result in more than 1 token-level conflict per line-level conflict. Our future work is to investigate a more complex model for dependence between token-level conflicts. We add a discussion of these statistics when we introduce Eq. (2).
>
> **Q: How do you split the train and test dataset (Table 4)? Is there a validation dataset?**
> A: The finetuning dataset is split into development and test set in the proportion 80/20 at random at file-level. The development set is further split into train and validation set in 80/20 proportion on conflict level.
>
> **Q: How do you deal with conflicts from more than 2 versions?**
> A: Conflicts involving more than 2 versions are not considered in this study. Our dataset only includes commits with 2 parents, that resulted in a merge conflict (we will correct the description in section 6 which says “at least two parents”).
> In practice, the strategy employed by existing systems when faced with a conflict of more than 2 versions is to decompose the problem into a series of pair-wise merges.  That is, merge the first two versions, then merge the result with the next version, etc.  Such a strategy could easily leverage our approach for merging two branches.
> Also, we note that such merges are quite rare.  In the Linux kernel (one of the most complex git repositories that exist) only 3% of merges include more than two branches (https://www.destroyallsoftware.com/blog/2017/the-biggest-and-weirdest-commits-in-linux-kernel-git-history).

---

> > ### Comment · Reviewer_k5J6 · 2021-11-19
> > **Discussion**
> >
> > Thanks the authors for the response.
> >
> > > Besides precision, the recall metric is crucial for real-world scenarios (a model that is correct 1 out of 1000 merges is not very interesting).
> >
> > Where do the `1` and `1000` numbers in your statement come from? I don't know whether you're implying recall or precision by saying that. Of course, the recall is important but the level of importance depends on applications. For conflict resolution, it would easier for a user to resolve the conflict themselves than trying to understand what's wrong with a resolution generated by an imprecise automated tool.
> >
> > That being said, I am not asking for a 100% precision, but there may be a better trade-off between precision and recall by calibrating your model.
> >
> > About the example, thanks for the derivation. How do you decide where and how many PAD tokens you need to add?

---

> > > ### Author Response · Authors · 2021-11-20
> > > **Re: Discussion**
> > >
> > > > Where do the `1` and `1000` numbers in your statement come from? I don't know whether you're implying recall or precision by saying that. Of course, the recall is important but the level of importance depends on applications. For conflict resolution, it would easier for a user to resolve the conflict themselves than trying to understand what's wrong with a resolution generated by an imprecise automated tool.
> > >
> > > We agree with the reviewer that both precision and recall metrics are important for the user experience. In the paper, we estimate the F-score, a harmonic mean of precision and recall, but other metrics could be measured during A/B tests or user studies: we elaborate on this below. The “1” and “1000” are metaphoric numbers, referring to a hypothetical tool with a low recall (but possibly high precision).
> > >
> > > > That being said, I am not asking for a 100% precision, but there may be a better trade-off between precision and recall by calibrating your model.
> > >
> > > An optimal trade-off between precision and recall would depend on various factors around the deployment and UX.
> > > However, there are several knobs one could use to increase the “online” precision of the tool (“online” as in IDE or a version control system):
> > > 1. Show top-K suggestions: more than 1 merge resolution suggestion could be shown e.g. as a dropdown list in an IDE (MergeBERT top-3 precision is 76%)
> > > 1. Introduce a log-probability threshold to further increase the precision
> > >
> > > We believe for actual deployment, an automatic resolution suggestion tool such as MergeBERT could be paired with oracles (require resolution to pass all unit tests in the project, or even formal verification of semantic conflict freedom) that ensure that the merged code is "correct". In such a case, the high recall from MergeBERT (75% top-3) becomes crucial to produce a semantically correct merge.
> > >
> > > > About the example, thanks for the derivation. How do you decide where and how many PAD tokens you need to add?
> > >
> > > First, we utilize difflib’s (https://docs.python.org/3/library/difflib.html) `SequenceMatcher` method to determine the longest contiguous matching subsequence following the Ratcliff-Obershelp algorithm. This algorithm will return a list of 5-tuples of the form `(tag, i1, i2, j1, j2)` describing how to turn one sequence into another. We then iterate over these 5-tuples, and, depending on the `tag`, decide on the placement of the `<pad>` tokens.
> > >
> > > Let us walk through the example. Given the tokenized sequences:
> > >
> > > `A_tokens`: `['--', 'num_cores', '=', '2']`
> > >
> > > `B_tokens`:  `['--', 'max_length', '=', '256', '--', 'num_cores', '=', '2']`
> > >
> > > A call to `SequenceMatcher` would give a list of following 5-tuples:
> > > 1. `('+', 0, 0, 0, 4)` # insert 4 tokens in the beginning of A
> > > 1.  `('=', 0, 4, 4, 8)` # unchanged
> > >
> > > We then iterate over the tuples to obtain the resulting diff token sequences for A and B (denoted as `result_A_tokens` and `result_B_tokens`), as well as the edit types.
> > >
> > > The first tuple gives ("+" tag, corresponds to insert action):
> > > ```
> > > edit_types +=  ["+"] * (j2 - j1)               # ['+', '+', '+', '+']
> > > result_A_tokens += ["<pad>"] * (j2 - j1)       # ['<pad>', '<pad>', '<pad>', '<pad>']
> > > result_B_tokens += B_tokens[j1:j2]             # ['--', 'max_length', '=', '256']
> > > ```
> > >
> > > Since the tag for the second tuple is “=” (equal), there are no padding tokens added:
> > > ```
> > > edit_types +=  ["="] * (j2 - j1)     # ['=', '=', '=', '=']
> > > result_A_tokens += A_tokens[i1:i2]   # ['--', 'num_cores', '=', '2']
> > > result_B_tokens += B_tokens[j1:j2]   # ['--', 'num_cores', '=', '2']
> > > ```
> > >
> > > For completeness, we also share the Python code from Yin et. al. We can add pseudocode in the appendix of the final version.
> > > ```
> > > def sequence_diff(base, reference):
> > >
> > >     matcher = difflib.SequenceMatcher()
> > >
> > >     base_tokens = []
> > >     reference_tokens = []
> > >     edit_types = []
> > >     matcher.set_seqs(base, reference)
> > >
> > >     for tag, i1, i2, j1, j2 in matcher.get_opcodes():
> > >         if tag == "=":
> > >             change_types.extend([0] * (i2 - i1))  # change type "="
> > >             base_tokens.extend(base[i1:i2])
> > >             reference_tokens.extend(reference[j1:j2])
> > >         elif tag == "-":
> > >             change_types.extend([1] * (i2 - i1))  # change type "-"
> > >             base_tokens.extend(base[i1:i2])
> > >             reference_tokens.extend(["<pad>"] * (i2 - i1))
> > >         elif tag == "+":
> > >             change_types.extend([2] * (j2 - j1))  # change type "+"
> > >             base_tokens.extend(["<pad>"] * (j2 - j1))
> > >             reference_tokens.extend(reference[j1:j2])
> > >         elif tag == "<->":
> > >             largest_span_size = max(i2 - i1, j2 - j1)
> > >             change_types.extend([3] * largest_span_size)  # change type "<->"
> > >             reference_tokens.extend(base[i1:i2] + ["<pad>"] * (largest_span_size - (i2 - i1)))
> > >             reference_tokens.extend(reference[j1:j2] + ["<pad>"] * (largest_span_size - (j2 - j1)))
> > >     assert len(change_types) == len(base_tokens) == len(reference_tokens)
> > >
> > >     return change_types, base_tokens, reference_tokens
> > > ```

---

### Official Review · Reviewer_vcTN · 2021-11-03

**Correctness:** 2
**Technical Novelty And Significance:** 2
**Empirical Novelty And Significance:** 2
**Recommendation:** 6
**Confidence:** 4

**Main Review:**

This is an important problem, and I'm excited to see more work in finding a neural solution to it.

However, this submission seems more like a refinement of DeepMerge than a novel contribution on its own. The data representation for conflicts is a generalization from DeepMerge, and the model architecture is a straighforward form of a Transformer-based classifier. However, the limited novelty might be offset by the positive progress on this important problem, modulo a number of questions below.

I wonder if there's room to incorporate the strategies of syntactic or lexical merging employed by FSTMerge and its variants and other non-neural baselines into the actions predicted by something like MergeBERT. It seems excessive to throw out the engineering that went into such non-neural tools, if they can be used for something more sophisticated than o, a, b, oa, ob, etc. Q1: I'd be curious if you think the strategies in those earlier tools are incompatible with an encoder like your Transformer-based encoder, too coarse grained, or perhaps just too complicated.

That said, I like the data-driven design of the task (e.g., the merge resolution "recipes" described in section 4). I would have liked to see where the 9 edit patterns came from though (presumably some analysis of real conflict resolutions?), since otherwise they appear arbitrary. Q2: are your classification labels arbitrary? Did you choose them by analyzing actual merge-conflict resolutions?

## Non-conflicting token edits

My biggest concern with your work is that you seem to treat token conflicts as independent of each other, but of course they are most certainly not, in general. For example, on page 3, with respect to Figure 1, I'd be careful about calling a token with no direct conflict a "non-conflicting edit" (e.g., the statement on `var` versus `let`). Consider the following diff:
```
<<<<<<<< A
a = func2(y, 10)
|||||||| O
a = func2(y, 9)
========
a = func3(y, 10, 12)
>>>>>>>> B
```
where `func2` is a function with arity 2, and `func3` is a function with arity 3. Here, although the `func2` to `func3` change on `B` is "non-conflicting", it is not independent of the change from `9)` to `10)` or `10, 12)`, since one keeps the arity at 2 and the other changes it to 3. Perhaps a model can learn these interdependencies but your model does not. I'd caution you against calling independent token changes "non-conflicting". Maybe there's a better motivating example than this one, or again, perhaps there's some analysis of real conflict resolutions showing that real conflicts are indeed token-independent.

## Evaluation

I don't quite understand your definition of precision and recall. Is precision perfect accuracy modulo whitespace per conflict region? What's recall? Why is it not 100%? Do you have a threshold on classification probability you're using that's causing you to lose some examples? It seems you have one interpretation of recall for diff3 (it cannot resolve the conflict) and possibly another for MergeBERT? Q3: Please explain your metrics with specificity.

It's not obvious how BLEU-4 is a relevant metric. In cases where token-level conflict resolution kicks in, A or B or even O probably share many n-grams with the resolution. What kind of solution does BLEU-4 allow you to score positively that would be genuinely acceptable for this task? An example would be helpful here. Q4: Please explain why BLEU-4 is a good metric for this task, perhaps with an example.

The zero-shot generalization to Scala is puzzling. Presumably the value proposition here is to argue that you don't have to pre-train for every language, and a single strong (potentially multilingual) model can do the job. But then I would want to see the comparison of pre-training/fine-tuning on Scala versus the zero-shot version with neither pre-training nor fine-tuning. In the absence of that, it's not clear what the headroom is for this language. Is the result good? Is it bad? Maybe pre-training and fine-tuning on Scala gives super high F1. Q5: How does MergeBERT do if you pre-train on Scala and fine-tune on Scala, and how does that compare to your zero-shot results?

I don't understand the point of Section 8.2. You already do MLM pre-training on your unilingual and multilingual pre-training datasets. What does this section add to what you already do? Q6: Please explain what research question Section 8.2 is answering, and what the answer is.



**Summary Of The Paper:**

This paper addresses merge-conflict resolution in source code repositories. Whereas prior neural work (DeepMerge) can only do line-level resolutions (i.e., predict a resolution that chooses some sequence of entire lines from the original file version or the two divergent branches), this work refines the task further towards token level resolutions, thereby covering more interesting cases. Whereas DeepMerge cast the task as a pointer-sequence prediction (predicted a sequence of lines from the input), MergeBERT casts the task as a classification problem: for each conflict instance (some token-sequence conflict), one of 9 recipes are chosen to merge (e.g., just the original tokens, just one of the branches, some concatenation of the three, and some substitution of tokens in the original by concatenations). Similarly to DeepMerge, the input is encoded as a combination of aligned token sequences of the original snapshots and the divergent branches, as well as an encoding of the token-wise edit "script" for the pairwise diffs (same, inserted, deleted, replaced). But instead of encoding this edit sequence explicitly, MergeBERT uses it as a token type embedding in a Transformer encoder, along with a positional embedding and the token embedding itself.

The Transformer encoder is pre-trained with the masked language model objective, before fine-tuning on merge-conflict resolutions. The resulting tool is shown to resolve more merge conflicts than prior work, and also to generalize to a language unseen during pre-training or fine-tuning.

**Summary Of The Review:**

Important problem, but there are many unanswered questions making the submission feel incomplete.

---

> ### Author Response · Authors · 2021-11-13
> **Response to reviewer vcTN**
>
> Thank you for your detailed review and thoughtful questions.
>
> **Q: My biggest concern with your work is that you seem to treat token conflicts as independent of each other, but of course they are most certainly not, in general.**
>
> A: The issue the reviewer raises demonstrates the difference between a semantic merge conflict vs a syntactic one. Indeed, the same argument could be made of the existing git diff3 algorithm: there is no guarantee that a clean git merge is semantically correct (where correct is defined in terms of all unit-tests, say, in the project)! In a semantic merge conflict, a syntactic merge is conflict free – but introduces a semantic difference (e.g,. compiler break, a bug, assertion violation, etc). This paper is focused on applying neural techniques to syntactic conflicts and demonstrates in most cases (82%), we match the user’s resolution. If we assume a user is unlikely to introduce a semantic conflict, then our results suggest the model is also able to mitigate semantic conflicts. That said, the reviewer is absolutely correct: the ultimate goal is a (semantic) conflict-free merge. This is a hard learning problem, and we see this paper as a step toward it. We will expound on this distinction in a future version.
>
> **Q: I'd be curious if you think the strategies in those earlier tools are incompatible with an encoder like your Transformer-based encoder, too coarse grained, or perhaps just too complicated.**
>
> A: Finding ways to incorporate semantic or syntactic strategies employed by FSTmerge (or other structured techniques) into neural models is a great research problem and part of the reason we hope this paper is accepted to ICLR! We note that our aim is to develop an approach that can be used for many languages; adding a new language requires only a tokenizer (which are freely available) and gathering additional data for that language.  Leveraging syntactic/semantic information similar to FSTMerge requires non-trivial engineering for each language.  For example each language has different semantics around properties of a correct merge such as identifying what structures can be reordered and which cannot. Further, in our experience FSTmerge leads to side effects (e.g. syntactically incorrect code) and cannot resolve various types of merges (for instance when conflict appears on a program statement).  This is a great research direction, but will require substantial innovation and engineering.
>
> **Q: are your classification labels arbitrary? Did you choose them by analyzing actual merge-conflict resolutions?**
> A: We used a data-driven approach to identify these 9 primitive merge resolution patterns based on analysis of the real-world merge conflict resolutions from GitHub. They cover almost all merges in practice.  Appendix 10.1 discusses the primitive merge patterns in more detail.
>
> **Q: Please explain your metrics with specificity.**
> A: Our definition of precision is standard: `Precision@K = correct@k/merged@k`, which is a ratio of correct merge resolutions predictions (true positives) divided by the total merged `TP/(TP+FP)`.
> The “positives” here are the cases for which the model was able to make a syntactically correct prediction (as determined by tree-sitter parser). Which is also why the fraction of merged programs is less than 100%. No log probability threshold is applied on neural model predictions. Recall is calculated as a ratio of correct merge resolution predictions divided by the total number of merge conflicts: `Recall@k = correct@k/total@k`. This is not a traditional definition, and we are happy to change the name to accuracy if you suggest to do so. We use this definition for recall instead of the standard `TP/(TP+FN)` because there is no easy way of estimating `FN`s, given all merges we consider had conflicts.  We will make this more clear in the final paper.
>
> **Q: Please explain why BLEU-4 is a good metric for this task, perhaps with an example.**
> A: We agree with the reviewer that BLEU and other n-gram based metrics are generally not a good choice for evaluating source code. We only consider BLEU-4 score as a secondary metric, and not include it in any tables or refer to it in supporting any conclusions. We will remove BLEU results to avoid the confusion.
>
> **Q: How does MergeBERT do if you pre-train on Scala and fine-tune on Scala, and how does that compare to your zero-shot results?**
> A: We have submitted an experiment you requested, and will add the number when available.
> The main motivation for zero-shot generalization experiments is to evaluate performance on relatively "unpopular” languages in Github (as in we do not have as much training data). Scala’s grammar and language features are somewhat like that of Java and C#. It provides language interoperability with Java so that libraries written in either language may be referenced directly in Scala or Java code. This partially explains a relatively good top-3 zero-shot performance.

---

### Author Response · Authors · 2021-11-16
**General Response**

**General response**

We thank all the reviewers for their time, feedback, and thoughtful comments. We respond here to questions raised in several reviews and include information about updates in the newest version. Responses to individual comments are posted as replies to each review.

**Syntactic vs Semantic merge conflicts and `diff3` (vcTN & k5j6)**

Both reviewers bring up excellent points about the `diff3` algorithm. Token-level `diff3` embodies the same intuition as a standard git line-level `diff3`; if one person changes a line, and the other does not, there is no __syntactic__ conflict, and thus `diff3` can merge that edit. Being a syntactic merge algorithm, `diff3` does not guarantee __semantic__ properties of a program. We see this work as a step toward the more general problem of building models that generate semantic free merges. As we point out to reviewer TUUK, we believe this more general semantic conflict-free merge resolution problem is a great task to drive modeling efforts of code because “… the dataset is plentiful (merges are common and easily mined from Github), the solutions are simpler than a traditional code generation task as most resolutions do not require generating new tokens, and finally, as the reviewer notes, “solving” this problem would have a positive impact in the daily lives of a huge number of people.”
Lastly, our data shows that in most cases (82\%), MergeBERT matches a user's resolution, which we assume does not have semantic issues.

**Impact of edit type information on the result (TUUK)**

We conducted an ablation study on the edit type embedding to understand the impact of edit-awareness of encoding on the model performance. As shown in Appendix 11.3, edit type embedding information improves MergeBERT’s F-score by 7\%.
We also included two ablation experiments to carry out a more precise comparison against the DeepMerge baseline, showing that MergeBERT outperforms DeepMerge with token diff by over 30\% F-score (7\% higher precision and 60\% higher recall), and to demonstrate that MergeBERT performance on line interleaving merge type is comparable to that on the entire test set.

---

### Decision · Program_Chairs · 2022-01-20

**Decision:**

Reject

**Comment:**

This paper is a fair effort, making some headway on a problem of practical importance.
There was some discussion of scoping and whether the contribution was Machine-Learning-y enough.
I'm kind of ambivalent on that particular question: I think the general rule is that the further out-of-scope the paper seems, the better the results need to be for people to overlook it.
I think in this case, unfortunately, even the two most positive reviewers did not evince enough excitement about this paper for it to get accepted in light of the scoping concerns.
Given the various constraints involved, I don't think I can recommend acceptance.

In order to get it accepted into a future conference I would recommend either:
a) Submit to a more Software-Engineering focused venue
b) Really shore up the evaluation such that the reviewers sympathetic to this kind of paper will find it unimpeachable and score it more generously.